# A New Layered Model on Emotional Intelligence

**DOI:** 10.3390/bs8050045

**Published:** 2018-05-02

**Authors:** Athanasios S. Drigas, Chara Papoutsi

**Affiliations:** 1Net Media Lab, Institute of Informatics & Telecommunications, National Centre for Scientific Research “Demokritos”, 15310 Agia Paraskevi, Greece; 2Net Media Lab, Institute of Informatics & Telecommunications, NCSR Demokritos, 15310 Agia Paraskevi, Greece; papoutsi.xara@yahoo.com

**Keywords:** intelligence, emotions, awareness, management, emotional intelligence

## Abstract

Emotional Intelligence (EI) has been an important and controversial topic during the last few decades. Its significance and its correlation with many domains of life has made it the subject of expert study. EI is the rudder for feeling, thinking, learning, problem-solving, and decision-making. In this article, we present an emotional–cognitive based approach to the process of gaining emotional intelligence and thus, we suggest a nine-layer pyramid of emotional intelligence and the gradual development to reach the top of EI.

## 1. Introduction

Many people misinterpret their own emotional reactions, fail to control emotional outbursts, or act strangely under various pressures, resulting in harmful consequences to themselves, others, and society. Other people have a greater ability to perform sophisticated information processing about emotions and emotion-relevant stimuli and to use this information as a guide for their own thoughts and behaviors and for others, in general [1].

Emotional intelligence (EI) is of great interest to scientists and researchers. Studies, from the past till today, continue to be made about the nature of emotional intelligence, its measurement, its structure, its positive and negative effects, and its relationship to many research fields [2,3,4,5,6,7,8]. Its influence on daily life in the short and long-term is important as well.

Intellectual ability is significant to succeed in everyday life within many different sectors [9,10,11,12]. Intelligence is an important aspect of the mind that includes a lot of cognitive abilities such as one’s abilities in logic, planning, problem-solving, adaptation, abstract thinking, understanding of ideas, language use, and learning [13,14]. However, there are some other important components that contribute to the aforementioned success including social capabilities, emotional adaptation, emotional sensitivity, empathy, practical intelligence, and incentives [15,16]. EI also focuses on the character and aspects of self-control, such as the ability to delay pleasures, the tolerance to frustrations, and the regulation of impulses (ego strength) [17]. Emotional intelligence also speaks to many areas of the psychological sciences—for example, the neuroscience of emotion, the theory of self-regulation, and metacognition—as well as the search for human cognitive abilities beyond what is traditionally known as academic intelligence [18,19].

In this paper, we are going to present the most discussed theories of intelligence, of emotions, and of emotional intelligence. We then present the construction of a 9-layer model (pyramid) of emotional intelligence which aims to show the levels a human must pass in order to reach the upper level of EI—emotional unity. The stratification of the pyramid of emotional intelligence is in tune with the pyramid of the functions of general intelligence [20].

## 2. Research Findings

### 2.1. Theories of Intelligence

The structure, nature, and characteristics of human intelligence have been discussed and have been the subject of debate since the time of Plato and Aristotle, at least a thousand years ago. Plato defined intelligence as a “learning tune” [21,22]. Under this concept, Plato and Aristotle put forth the three components of mind and soul: intellect, sentiment, and will [23]. The word “intelligence” comes from two Latin words: intellegentia and ingenium. The first word, considered in the way Cicero used the term, means “understanding” and “knowledge”. The second word means “natural predisposition” or “ability” [24]. 

At various points in recent history, researchers have proposed different definitions to explain the nature of intelligence [22]. The following are some of the most important theories of intelligence that have emerged over the last 100 years.

Charles Spearman [25] developed the theory of the two factors of intelligence using data factor analysis (a statistical method) to show that the positive correlations between mental examinations resulted from a common underlying agent. Spearman suggested that the two-factor theory had two components. The first was general intelligence, *g*, which affected one’s performance in all mental tasks and supported all intellectual tasks and intellectual abilities [25,26]. Spearman believed that the results in all trials correlated positively, underlying the importance of general intelligence [25,27]. The second agent Spearman found was the specific factor, *s*. The specific factor was associated with any unique capabilities that a particular test required, so it differed from test to test [25,26]. Regarding *g*, Spearman saw that individuals had more or less general intelligence, while *s* varied from person to person in a job [28]. Spearman and his followers gave much more importance to general intelligence than to the specific agent [25,29].

In 1938, American psychologist Louis L. Thurstone suggested that intelligence was not a general factor, but a small set of independent factors that were of equal importance. Thurstone formulated a model of intelligence that centered on “Primary Mental Abilities” (PMAs), which were independent groups of intelligence that different individuals possessed in varying degrees. To detect these abilities, Thurstone and his wife, Thelma, thought up of a total of 56 exams. They passed the test bundle to 240 students and analyzed the scores obtained from the tests with new methods of Thurstone’s method of analysis. Thurstone recognized seven primary cognitive abilities: (1) verbal understanding, the ability to understand the notions of words; (2) verbal flexibility, the speed with which verbal material is handled, such as in the production of rhymes; (3) number, the arithmetic capacity; (4) memory, the ability to remember words, letters, numbers, and images; (5) perceptual speed, the ability to quickly discern and distinguish visual details, and the ability to perceive the similarities and the differences between displayed objects; (6) inductive reasoning, the extraction of general ideas and rules from specific information; and (7) spatial visualization, the ability to visualize with the mind and handle objects in three dimensions [30,31].

Joy Paul Guilford extended Thurstone’s work and devoted his life to create the model for the structure of intelligence. SI (Structure of Intellect theory, 1955) contains three dimensions: thought functions, thought content, and thought products. Guilford described 120 different kinds of intelligence and 150 possible combinations. He also discovered the important distinction between convergent and divergent thought. The convergent ability results in how well one follows the instructions, adheres to rules, and tries. The divergent ability decreases depending on whether or not one follows the instructions or if one has a lot of questions, and it usually means that one is doing the standard tests badly [32,33].

The Cattell-Horn Gf-Gc and the Carroll Three-Stratum models are consensual psychometric models that help us understand the construction of human intelligence. They apply new methods of analysis and according to these analyses, there are two basic types of general intelligence: fluid intelligence (gf) and crystallized intelligence (gc). Fluid intelligence represents the biological basis of intelligence. How fast someone thinks and how well they remember are elements of fluid intelligence. These figures increase in adulthood but as we grow older they decrease. Fluid intelligence enables a person to think and act quickly, to solve new problems, and to encode short-term memories. Crystallized intelligence, on the other hand, is the knowledge and skills acquired through the learning process and through experience. Crystallized abilities come from learning and reading and are reflected in knowledge trials, general information, language use (vocabulary), and a wide variety of skills. As long as learning opportunities are available, crystallized intelligence may increase indefinitely during a person’s life [14,34].

In the 1980s, the American psychologist Robert Sternberg proposed an intelligence theory with which he tried to extend the traditional notion of intelligence. Sternberg observed that the mental tests that people are subjected to for various intelligence measurements are often inaccurate and sometimes inadequate to predict the actual performance or success. There are people who do well on the tests but not so well in real situations. Likewise, the opposite occurred as well. According to Sternberg’s triarchic (three-part) theory of intelligence, intelligence consists of three main parts: analytical intelligence, creative intelligence, and practical intelligence. Analytical intelligence refers to problem-solving skills, creative intelligence includes the ability to handle new situations using past experiences and current skills, and practical intelligence refers to the ability to adapt to new situations and environments [35,36].

In 1983, psychologist Howard Gardner introduced his theory of Multiple Intelligences (MI), which, at that time, was a fundamental issue in education and a controversial topic among psychologists. According to Gardner, the notion of intelligence as defined through the various mental tests was limited and did not depict the real dimensions of intelligence nor all the areas in which a person can excel and succeed. Gardner argued that there is not only one kind of general intelligence, but rather that there are multiple intelligences and each one is part of an independent system in the brain. The theory outlines eight types of “smart”: Linguistic intelligence (“word smart”), Logical–mathematical intelligence (“number/reasoning smart”), Spatial intelligence (“picture smart”), Bodily–Kinesthetic intelligence (“body smart”), Musical intelligence (“music smart”), Interpersonal intelligence (“people smart”), Intrapersonal intelligence (“self smart”), and Naturalist intelligence (“nature smart”) [37,38].

### 2.2. Emotions

According to Darwin, all people, irrespective of their race or culture, express emotions using their face and body with a similar way as part of our evolutionary heritage [39,40]. Emotion is often defined as a complex feeling which results in physical and psychological changes affecting thought and behavior. Emotions include feeling, thought, nervous system activation, physiological changes, and behavioral changes such as facial expressions. Emotions seem to dominate many aspects of our lives as we have to recognize and to respond to important events related to survival and/or the maintenance of prosperity and, therefore, emotions serve various functions [41]. Emotions are also recognized as one of the three or four fundamental categories of mental operations. These categories include motivation, emotion, cognition, and consciousness [42]. Most major theories of emotion agree that cognitive processes are a very important source of emotions and that feelings comprise a powerful motivational system that significantly influences perception, cognition, confrontation, and creativity [43]. Researchers have been studying how and why people feel emotion for a long time so various theories have been proposed. These include evolutionary theories [44,45], the James-Lange Theory [46,47], the Cannon-Bard Theory [48], Schacter and Singer’s two-factor theory [49,50], and cognitive appraisal [51]. 

### 2.3. Emotional Intelligence

Anyone can become angry-that is easy. But to be angry with the right person, to the right degree, at the right time, for the right purpose, and in the right way-this is not easy.—Aristotle, The Nicomachean Ethics

Thorough research has indicated the important role that emotions play in our lives in many fields [52,53,54,55]. Researchers have found that Emotional Intelligence is equal to or sometimes much more important than I.Q [56,57,58,59,60]. Emotion and intelligence are heavily linked [61,62,63]. If you are aware of your own and others’ feelings, this will help you manage behaviors and relationships and predict success in many sectors [64,65,66].

Emotional Intelligence is the ability to identify, understand, and use emotions positively to manage anxiety, communicate well, empathize, overcome issues, solve problems, and manage conflicts. According to the Ability EI model, it is the perception, evaluation, and management of emotions in yourself and others [67]. Emotional Intelligence (EI), or the ability to perceive, use, understand, and regulate emotions, is a relatively new concept that attempts to connect both emotion and cognition [68].

Emotional Intelligence first appeared in the concept of Thorndike’s “social intelligence” in 1920 and later from the psychologist Howard Gardner who, in 1983, recommended the theory of multiple intelligence, arguing that intelligence includes eight forms. American psychologists Peter Salovey and John Mayer, who together introduced the concept in 1990 [69], define emotional intelligence “as the ability to monitor one’s own and other’s emotions, to discriminate among them, and to use the information to guide one’s thinking and actions”. People who have developed their emotional intelligence have the ability to use their emotions to direct thoughts and behavior and to understand their own feelings and others’ feelings accurately. Daniel Goleman, an American writer, psychologist, and science journalist, disclosed the EI concept in his book named “Emotional Intelligence” [58,59,60]. He extended the concept to include general social competence. Goleman suggested that EI is indispensable for the success of one’s life.

Mayer and Salovey suggested that EI is a cognitive ability, which is separate but also associated with general intelligence. Specifically, Mayer, Salovey, Caruso, and Sitarenios [70] suggested that emotional intelligence consists of four skill dimensions: (1) perceiving emotion (i.e., the ability to detect emotions in faces, pictures, music, etc.); (2) facilitating thought with emotion (i.e., the ability to harness emotional information in one’s thinking); (3) understanding emotions (i.e., the ability to understand emotional information); and (4) managing emotions (i.e., the ability to manage emotions for personal and interpersonal development). These skills are arranged hierarchically so that the perceptual emotion has a key role facilitating thinking, understanding emotions, and managing emotions. These branches are arising from higher order basic skills, which are evolved as a person matures [67,71].

According to Bar-On emotional-social intelligence is composed of emotional and social abilities, skills and facilitators. All these elements are interrelated and work together. They play a key role in how effectively we understand ourselves and others, how easily we express ourselves, but also in how we deal with daily demands [72].

Daniel Goleman (1998) defines Emotional Intelligence/Quotient as the ability to recognize our own feelings and those of others, to motivate ourselves, and to handle our emotions well to have the best for ourselves and for our relationships. Emotional Intelligence describes capacities different from, but supplementary to, academic intelligence. The same author introduced the concept of emotional intelligence and pointed out that it is composed of twenty-five elements which were subsequently compiled into five clusters: Self Awareness, Self-Regulation, Motivation, Empathy, and Social Skills [61,73]. 

Petrides and Furnham (2001) developed the Trait Emotional Intelligence model which is a combination of emotionally-related self-perceived abilities and moods that are found at the lowest levels of personality hierarchy and are evaluated through questionnaires and rating scales [74]. The trait EI essentially concerns our perceptions of our inner emotional world. An alternative tag for the same construct is trait emotional self-efficacy. People with high EI rankings believe that they are “in touch” with their feelings and can regulate them in a way that promotes prosperity. These people may enjoy higher levels of happiness. The trait EI feature sampling domain aims to provide complete coverage of emotional aspects of personality. Trait EI rejects the idea that emotions can be artificially objectified in order to be graded accurately along the IQ lines [75]. The adult sampling domain of trait EI contains 15 facets: Adaptability, Assertiveness, Emotion perception (self and others), Emotion expression, Emotion management (others’), Emotion regulation, Impulsiveness (low), Relationships, Self-esteem, Self-motivation, Social awareness, Stress management, Trait empathy, Trait happiness, and Trait optimism [76].

Research on emotional intelligence has been divided into two distinct areas of perspectives in terms of conceptualizing emotional competencies and their measurements. There is the ability EI model [77] and the trait EI [74]. Research evidence has consistently supported this distinction by revealing low correlations between the two [64,78,79,80,81]. 

EI refers to a set of emotional abilities that are supposed to foretell success in the real world above and beyond general intelligence [82,83]. Some findings have shown that high EI leads to better social relationships for children [84], better social relations for adults [85], and more positive perception of individuals from others [85]. High EI appears to influence familial relationships, intimate relationships [86], and academic achievement positively [87,88]. Furthermore, EI consistently seems to predict better social relations during work performance and in negotiations [89,90] and a better psychological well-being [91].

## 3. The Pyramid of Emotional Intelligence: The Nine-Layer Model

Τaking into consideration all the theories of the past concerning pyramids and layer models dealing with EI, we analyze the levels of our pyramid step by step (Figure 1), their characteristics, and the course of their development so as to conquer the upper levels, transcendence and emotional unity, as well as pointing out the significance of EI. Our model includes features from both constructions (the Ability EI and the Trait EI model) in a more hierarchical structure. The ability level refers to awareness (self and social) and to management. The level of trait refers to the mood associated with emotions and the tendency to behave in a certain way in emotional states considering other important elements that this construction includes as well. The EI pyramid is also based on the concepts of intrapersonal and interpersonal intelligences of Gardner [92,93].

### 3.1. Emotional Stimuli

Every day we receive a lot of information-stimuli from our environment. We need to incorporate this information and the various stimuli into categories because they help us to understand the world and the people that surround us better [94]. The direct stimulus of emotions is the result of the sensorial stimulus processing by the cognitive mechanisms [95,96,97]. When an event occurs, sensorial stimuli are received by the agent. The cognitive mechanisms process this stimulus and produce the emotional stimuli for each of the emotions that will be affected [98]. Emotional stimuli are processed by a cognitive mechanism that determines what emotion to feel and subsequently produce an emotional reaction which may influence the occurrence of the behavior. Emotional stimuli are generally prioritized in perception, are detected more quickly, and gain access to conscious awareness [99,100]. The emotional stimuli constitute the base of the pyramid of emotional intelligence pointing to the upper levels of it.

### 3.2. Emotion Recognition

The next level of the pyramid after the emotional stimuli is the recognition of emotions simultaneously expressed at times. Accuracy is higher when emotions are both expressed and recognized. Emotion recognition includes the ability to accurately decode the expressions of others’ feelings, usually transmitted through non-verbal channels (i.e., the face, body, and voice). This ability is positively linked to social ability and interaction, as non-verbal behavior is a reliable source of information on the emotional states of others [101]. Elfenbein and Ambady commented that emotion recognition is the most “reliably validated component of emotional intelligence” linked to a variety of positive organizational outcomes [102]. The ability to express and recognize emotions in others is an important part of the daily human interaction and interpersonal relationships as it is a representation of a critical component of human socio-cognitive capacities [103].

### 3.3. Self-Awareness

Socrates mentions in his guiding principle, “know thyself”. Aristotle also mentioned “knowing yourself is the beginning of all wisdom”. These two ancient Greek aphorisms encompass the concept of self-awareness, a cognitive capacity, which is the following step in our pyramid after having conquered the previous two. Self-Awareness is having a clear perception of your personality, including your strengths, weaknesses, thoughts, beliefs, motives, and feelings [104]. As you develop self-awareness, you are able to change your thoughts which, in turn, allow you to change your emotions and eventually change your actions. Crisp and Turner [105] described self-awareness as a psychological situation in which people know their traits, feelings, and behaviors. Alternatively, it can be defined as the realization of oneself as an individual entity. Developing self-awareness is the first step to develop your EI. The lack of self-awareness in terms of understanding ourselves and having a sense of ourselves that has roots in our own values impedes our ability to self-manage and it is difficult, if not impossible, to know and to respond to the others’ feelings [61]. Daniel Goleman [106,107] recognized self-awareness as emotional consciousness, accurate self-esteem, and self-confidence. Knowing yourself means having the ability to understand your feelings, having an accurate self-assessment of your own strengths and weaknesses, and showing self-confidence. According to Goleman, self-awareness must be ahead of social awareness, self-management, and relationship management which are important factors of EI.

### 3.4. Self-Management

Once you have clarified your emotions and the way they can affect the situations and other people, you are ready to move to the EQ area of self-management. Self-management allows you to control your reactions so that you are not driven by impulsive behaviors and feelings. With self-management, you become more flexible, more extroverted, and receptive, and at the same time less critical on situations and less reactionary to people’s attitudes. Moreover, you know more about what to do. When you have recognized your feelings and have accepted them, you are then able to manage them much better. The more you learn on the way to manage your emotions, the greater your ability will be to articulate them in a productive way when need be [108]. This does not mean that you must crush your negative emotions, but if you realize them, you can amend your behavior and make small or big changes to the way you react and manage your feelings even if the latter is negative. The second emotional intelligence (EQ) quadrant of self-management consists of nine key components: (1) emotional self-control; (2) integrity; (3) innovation and creativity; (4) initiative and prejudice to action; (5) resilience; (6) achievement guide; (7) stress management; (8) realistic optimism and (9) intentionality [80,106,107,109].

### 3.5. Social Awareness—Empathy—The Discrimination of Emotions

Since you have cultivated the ability to understand and control your own emotions, you are ready to move on to the next step of recognizing and understanding the emotions of people around you. Self-Management is a prerequisite for Social-Awareness. It is an expansion of your emotional awareness. Social Awareness refers to the way people handle relationships and awareness of others’ feelings, needs, and concerns [110]. The Social Awareness cluster contains three competencies: Empathy, Organizational Awareness, Service Orientation [107]. Being socially aware means that you understand how you react to different social situations, and effectively modify your interactions with other people so that you achieve the best results. Empathy is the most important and essential EQ component of social awareness and is directly related to self-awareness. It is the ability to put oneself in another’s place (or “shoes”), to understand him as a person, to feel him and to take into account this perspective related to this person or with any person at a time. With empathy, we can understand the feelings and thoughts of others from their own perspective and have an active role in their concerns [111]. The net result of social awareness is the ongoing development of social skills and a personal continuous improvement process [107,112,113]. Discrimination of emotions belongs to that level of the pyramid because it is a rather intellectual ability that gives people the capacity to discriminate with accuracy between different emotions and label them appropriately. The latter in relation to the other cognitive functions contributes to guide thinking and behavior [77].

### 3.6. Social Skills—Expertise

After having developed social awareness, the next level in the pyramid of emotional intelligence that helps raising our EQ is that of social skills. In emotional intelligence, the term social skills refers to the skills needed to handle and influence other people’s emotions effectively to manage interactions successfully. These abilities range from being able to tune into another person’s feelings and understand how they feel and think about things, to be a great collaborator and team player, to expertise at emotions of others and at negotiations. It is all about the ability to get the best out of others, to inspire and to influence them, to communicate and to build bonds with them, and to help them change, grow, develop, and resolve conflict [114,115,116]. Social skills under the branch of emotional intelligence can include Influence, Leadership, Developing Others, Communication, Change Catalyst, Conflict Management, Building Bonds, Teamwork, and Collaboration [61]. Expertise in emotions could be characterized as the ability to increase sensitivity to emotional parameters and the ability not only to accurately determine the relevance of emotional dynamics to negotiation but also the ability to strategically expose the emotions of the individual and respond to emotions stemming from others [117].

### 3.7. Self-Actualization—Universality of Emotions

As soon as all six of these levels have been met, the individual has reached the top of Maslow’s hierarchy of needs; Self-Actualization. Every person is capable and must have the will to move up to the level of self-actualization. Self-Actualization, according to Maslow [118,119,120], is the realization of personal potential, self-fulfillment, pursuing personal development and peak experiences. It is important to note that self-actualization is a continual process of becoming, rather than a perfect state one reaches such as a ‘happy ever after” [121]. Carl Rogers [122,123] also created a theory that included a “growth potential” whose purpose was to incorporate in the same way the “real self” and the “ideal self”, thereby cultivating the appearance of the “fully functioning person”. Self-actualization is one of the most important EI skills. It is a measure of your sense that you have a substantial personal commitment to life and that you are offering the gifts to your world that are most important for you. Reuven Bar-On [124] illustrates the close relationship between emotional intelligence and self-actualization. His research led him to conclude that “you can actualize your potential capacity for personal growth only after you are socially and emotionally effective in meeting your needs and dealing with life in general”. Self-actualizers feel empathy and kinship towards humanity as a whole and therefore, that cultivates the universality of emotions, so that those they have emotional intelligence in one culture probably have emotional intelligence in another culture too and they have the ability to understand the difference of emotions and their meanings despite the fact that sometimes emotions are culturally dependent [125,126].

### 3.8. Transcendence

Maslow also proposed that people who have reached self-actualization will sometimes experience a state he referred to as “transcendence”. In the level of Transcendence, one helps others to self-actualize, find self-fulfillment, and realize their potential [127,128]. The emotional quotient is strong and those who have reached that level try to help other people understand and manage their own and others’ emotions too. Transcendence refers to the much higher and more comprehensive or holistic levels of human consciousness, by behaving and associating, as ends rather than as means, to ourselves, to important others, to human in general, to other species, to nature, and to the world [129]. Transcendence is strongly correlated with self-esteem, emotional well-being and global empathy. Self-transcendence is the experience of seeing yourself and the world in a way that is not impeded by the limits of one’s ego identity. It involves an increased sense of meaning and relevance to others and to the world [130,131]. In his perception of transcendence Plato affirmed the existence of absolute goodness that he characterized as something that cannot be described and it is only known through intuition. His ideas are divine objects that are transcendent of the world. Plato also speaks of gods, of God, of the cosmos, of the human soul, and of that which is real in material things as transcendental [132]. Self-transcendence can be expressed in various ways, behaviors and perspectives like the exchange of wisdom and emotions with others, the integration of physical/natural changes of aging, the acceptance of death as part of life, the interest in helping others and learning about the world, the ability to leave your losses behind, and the finding of spiritual significance in life [133].

### 3.9. Emotional Unity

Emotional unity is the final level in our pyramid of emotional intelligence. It is an intentionally positive oriented dynamic, in a sense that it aims towards reaching and keeping a dominance of emotions, which inform the subject that he or she is controlling the situation or the setting in an accepted shape. This reached level of emotional unity in the subject can be interpreted as an outcome of emotional intelligence [134]. The emotional unity is an internal harmony. In emotional unity one feels intense joy, peace, prosperity, and a consciousness of ultimate truth and the unity of all things. In a symbiotic world, what you do for yourself, you ultimately do for another. It all starts with our love for ourselves, so that we can then channel this important feeling to everything that exists around us [135]. Not only in human beings, but also in animals, plants, oceans, rocks, and so forth. All it takes is to see the spark of life and miracle in everything and be more optimistic. The point is that somehow, we are all interconnected, and the more we delve deeper our heart and follow it, the less likely it will be for us to do things that can harm others or the planet in general [136]. The others are not separate from us. Emotional unity emanates humility and empathy that bears with the imperfections of the other. Plato in Parmenides also talks about unity [137], Being, and One. As Parmenides writes: “Being is ungenerated and indestructible, whole, of one kind and unwavering, and complete. Nor was it, nor will it be, since now it is, all together, one, continuous…” [138,139].

## 4. Cognitive and Metacognitive Processes in the Emotional Intelligence Pyramid

Cognition encompasses processes such as attention, memory, evaluation, problem-solving language, and perception [140,141]. Cognitive processes use existing knowledge and generate new knowledge. Metacognition is defined as the ability to monitor and reflect upon one’s own performance and capabilities [142,143]. It is the ability of individuals to know their own cognitive functions in order to monitor and to control their learning process [144,145]. The idea of meta-cognition relies on the distinction between two types of cognitions: primary and secondary [146]. Metacognition includes a variety of elements and skills such as Metamemory, Self-Awareness, Self-Regulation, and Self-Monitoring [144,147].

Metacognition in Emotional Intelligence means that an individual perceives his/her emotional skills [148,149]. Its processes involve emotional-cognitive strategies such as awareness, monitoring, and self-regulation [150]. Apart from the primary emotion, a person can experience direct thoughts that accompany this emotion as people may have additional cognitive functions that monitor a given emotional situation [151], they may evaluate the relationship between emotion and judgment [152], and they may try to manage their emotional reaction [153] for the improvement of their own personality and that will motivate them to help other people for better interpersonal interactions. Applying the meta-knowledge to socio-emotional contexts should lead to the opportunity to learn to correct one’s emotional errors and to promote the future possibility of a proper response to the situation while maintaining and cultivating the relationship [154].

In the pyramid of Emotional Intelligence, to move from one layer to another, cognitive and metacognitive processes are occurred (Figure 2).

## 5. Discussion & Conclusions

Emotional Intelligence is a very important concept that has come back to the fore in the last decades and has been the subject of serious discussions and studies by many experts. The importance of general intelligence is neither underestimated nor changed, and this has been proven through many surveys and studies.

On the other hand, however, we must also give emotional intelligence the place it deserves. The cultivation of emotional intelligence can contribute to and provide many positive benefits to people’s lives in accordance with studies, surveys, and with what has been already mentioned. When it comes to happiness and success in life, emotional intelligence (EQ) matters just as much as intellectual ability (IQ) [60]. Furthermore, it should be noted that despite the various discussions about emotional intelligence, studies have shown that emotional abilities that make up emotional intelligence are very important for the personal and social functioning of humans [83]. A core network of brain regions such as the amygdala and ventromedial prefrontal cortex is the key to a range of emotional abilities and plays a crucial role for human lesions [155]. Specific Emotional Intelligence components (Understanding Emotions and Managing Emotions) are directly related to the structural microarchitecture of major axial pathways [156].

With emotional intelligence you acknowledge, accept, and control your emotions and emotional reactions as well as those of other people. You learn about yourself and move on to the understanding of other people’s self. You learn to coexist better, which is very important since we are not alone in this world and because when we want to advance ourselves, and society as a whole, there must be cooperation and harmony. With emotional intelligence, you learn to insist, to control your impulses, to survive despite adversities and difficulties, to hope for and to have empathy. Emotional Intelligence provides you with a better inner world to cope with the outside world according to Trait EI [157]. It involves and engages higher cognitive functions such as attention, memory, regulation, reasoning, awareness, monitoring, and decision-making. The results show that negative mood and anticipated fear are two factors of the relationship between trait EI and risk-taking in decision-making processes among adults [158]. Research has also shown this positive correlation between emotional intelligence and cognitive processes and this demonstrates the important role that emotional intelligence plays with emotion and cognition, thus, empowering individuals and their personality and benefitting the whole society [159,160,161,162,163,164].

Αs we rise through the levels of the pyramid of emotional intelligence that we have presented, we step closer to its development to the fullest extent, to the universality of emotions, to emotional unity. The human being is good at trying to reach the last level of the pyramid because at each level he cultivates significant emotional, cognitive, and metacognitive skills that are important resources for the successes in one’s personal life, professional life, interpersonal relationships, and in life in general.

Emotional intelligence is a skill that can be learned and developed [165,166]. The model of emotional intelligence has been created with a better distinct classification. It is a more structured evaluation and intervention model with hierarchical levels to indicate each level of emotional intelligence that everyone is at and with operating procedures to contribute to the strengthening of that level and progressive development of the individual to the next levels of emotional intelligence. It is a methodology for the further development and evolution of the individual. This model can have practical applications as an evaluation, assessment, and training tool in any aspect of life such as interpersonal relationships, work, health, special education, general education, and academic success. Researchers claim that an emotional mind is important for a good life as much as an intelligent mind and, in certain cases, it matters more [167]. The ultimate goal should be to develop Emotional Intelligence, do further research on the benefits of such an important capacity and the correlations between the layered Emotional Intelligence model and other variables. 

In this paper, we presented the pyramid of Emotional Intelligence as an attempt to create a new layer model based on emotional, cognitive, and metacognitive skills. In essence, each higher level of the pyramid is an improvement toward one’s personal growth and a higher state of self-regulation, self-organization, awareness, consciousness, attention, and motivation. 

## Figures and Tables

**Figure 1 behavsci-08-00045-f001:**
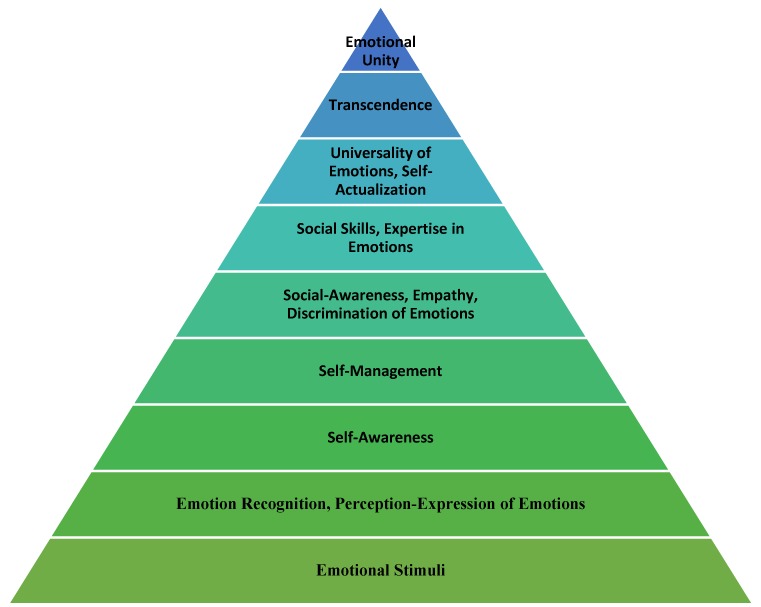
The emotional intelligence pyramid (9-layer model).

**Figure 2 behavsci-08-00045-f002:**
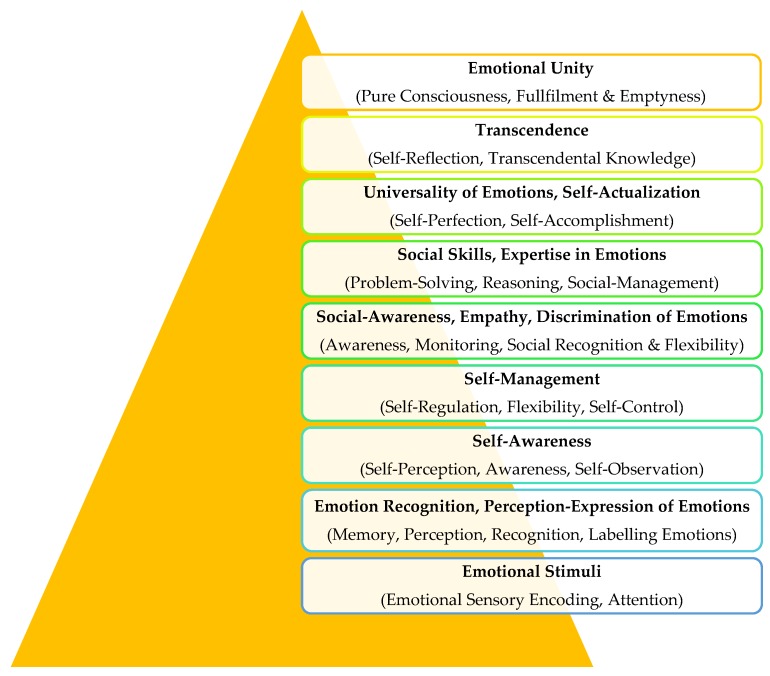
The cognitive and metacognitive processes to move from a layer to another.

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
