# Peer review of "A New Layered Model on Emotional Intelligence"

_behavsci, 2018, doi:10.3390/bs8050045_

Round 1
Reviewer 1 Report
The manuscript includes a brief introduction on the psychological constructs of intelligence and emotional intelligence and the related primary theories and models. Besides discussing the relevance of developing good levels of emotional intelligence and related affective/social skills, the authors present a nine-layers hierarchical descriptive model of emotional intelligence. The model grounds on and summarizes various ideas and constructs from many philosophical/psychological theories of emotions and affective intelligence.
The topic is interesting and has practical implications in many fields.
The theoretical model and related remarks are clearly presented, even if the discussion would have benefitted from additional critical discussion and from specific discussion of neural correlates of the skills and levels that are taken into consideration.
A few minor stylistic, grammar, and language issues are present (see below).
Minor:
1) We recommend language revision – e.g. p.5 ll.221-223 “The EI pyramid also encompasses the concepts of intrapersonal and interpersonal intelligences postulated by Gardner”
2) Pay attention to typos, many of them are still present in the manuscript - e.g. p.1, l.32 "ego stenght"; p.5 l.195 “(Bar-On, 2006 (Bar-On, 2006)”
3) Pay attention to citation and reference editing norms – e.g. p.9 l.403 “(Vardhan; Goleman, 1995)”, Vardhan’s work has no related year; p.15 l. 679, the reference “Vardhan, J. V. Emotional Intelligence (EQ)” lacks of many pieces of information
4) Pay attention to APA norms for quotations – p.9 ll.387-389 ““Being is ungenerated and indestructible, whole, of one kind 387 and unwavering, and complete. Nor was it, nor will it be, since now it is, all together, one, 388 continuous... (Cornford, 1939; Burnet, 1908).””
Author Response
We would like to thank the reviewers for careful and thorough reading of this manuscript and for the thoughtful comments and constructive suggestions, which help to improve the quality of this manuscript. With this cover letter, we will submit the revised manuscript (behavsci-270715) entitled, “A New Layered Model on Emotional Intelligence” for publication in Behavioral Sciences. Based the comments from the reviewers, we have made changes of the manuscript, which are detailed below (the reviewer’s comments are in italics). Changes made in the manuscript are marked using track changes and hope to meet with your approval.
REVIEWER #1:
Major changes:
1)The theoretical model and related remarks are clearly presented, even if the discussion would have benefitted from additional critical discussion and from specific discussion of neural correlates of the skills and levels that are taken into consideration.
Reply: We have included the following sentences to illustrate the interesting perspective you mentioned in the Discussion-Conclusions (lines 428-438):
“Furthermore, it should be noted that despite the various discussions about emotional intelligence, studies have shown that emotional abilities that make up emotional intelligence are very important for human personal and social functioning in terms of human lesions (Hogeveen et al. 2016;). A core network of brain regions such us amygdala and ventromedial prefrontal cortex is the key to a range of emotional abilities and plays a crucial role for human lesions (Operskalski et al. 2015). Specific Emotional Intelligence components (Understanding Emotions and Managing Emotions) are directly related to the structural microarchitecture of major axial pathways. These findings suggest that specific components of EI (Understanding Emotions and Managing Emotions) are immediately related to the structural microarchitecture of major axonal pathways (Pisner et al. 2017).”
Minor changes:
1) We recommend language revision – e.g. p.5 ll.221-223 “The EI pyramid also encompasses the concepts of intrapersonal and interpersonal intelligences postulated by Gardner”
Reply: This sentence has been revised to: “The EI pyramid is also based on the concepts of intrapersonal and interpersonal intelligences of Gardner (Gardner, 2000; 2011).
2) Pay attention to typos, many of them are still present in the manuscript - e.g. p.1, l.32 "ego stenght"; p.5 l.195 “(Bar-On, 2006 (Bar-On, 2006)”
Reply: Typos mentioned were corrected.
3) Pay attention to citation and reference editing norms – e.g. p.9 l.403 “(Vardhan; Goleman, 1995)”, Vardhan’s work has no related year; p.15 l. 679, the reference “Vardhan, J. V. Emotional Intelligence (EQ)” lacks of many pieces of information
Reply: The website reference was deleted because there was no related year as you correctly mentioned.
4) Pay attention to APA norms for quotations – p.9 ll.387-389 ““Being is ungenerated and indestructible, whole, of one kind 387 and unwavering, and complete. Nor was it, nor will it be, since now it is, all together, one, 388 continuous... (Cornford, 1939; Burnet, 1908).””
Reply: The quotation was corrected.
Reviewer 2 Report
Thank you for the opportunity to review the paper titled “A New Layered Model on Emotional Intelligence” The manuscript under review shows a new theoretical model of EI. The topic is relevant and I think that this field of research deserves more attention.
Here follow my concerns:
I went throughout the manuscript but I could not see any reference linking the proposed model to trait emotional intelligence construct postulated from Petrides and colleagues. I really believe that authors should include Petrides perspective into their manuscript and discuss possible links among these two models.
Authors mention a potential link between EI and decision-making processes but it seems unexplored. They should add some references concerning literature about EI and decision-making processes (e.g., Panno et al., https://doi.org/10.1027/1864-9335/a000247; Sevdalis et al., https://doi.org/10.1016/j.paid.2006.10.012 ). As the DM processes are higher-order cognitive factors that strongly influence daily life outcomes, the manuscript would benefit from adding a paragraph linking these aspects.
Finally, the manuscript would benefit from adding a section describing potential applied interventions of the model. Moreover, some suggestions on how people EI could be enhanced based on this model are needed. The discussion section of the manuscript reports a very small number of prompts and suggestions about applied research and future studies investigating EI phenomenon.
In conclusion, I think authors should paid less attention to old theories of EI, and instead, paid more attention to recent theories shedding light on EI construct.
Minor comments:
Row 195: there is a typo.
Row 272: “clearly clarified” does not sound good.
Rows 411/412: a reference is missing.
Row 419: see above.
Author Response
We would like to thank the reviewers for careful and thorough reading of this manuscript and for the thoughtful comments and constructive suggestions, which help to improve the quality of this manuscript. With this cover letter, we will submit the revised manuscript (behavsci-270715) entitled, “A New Layered Model on Emotional Intelligence” for publication in Behavioral Sciences. Based the comments from the reviewers, we have made changes of the manuscript, which are detailed below (the reviewer’s comments are in italics). Changes made in the manuscript are marked using track changes and hope to meet with your approval.
REVIEWER #2:
Major changes:
1)I went throughout the manuscript, but I could not see any reference linking the proposed model to trait emotional intelligence construct postulated from Petrides and colleagues. I really believe that authors should include Petrides perspective into their manuscript and discuss possible links among these two models.
Reply: We agree with the reviewer and have added the following paragraphs in chapter 3.3. Emotional Intelligence (lines 206-225):
“Petrides and Furnham (2001) developed the Trait Emotional Intelligence model which is a combination of emotionally-related self-perceived abilities and moods that are at the lowest levels of personality hierarchy evaluated through questionnaires and rating scales (Petrides & Furnham, 2001). The trait EI essentially concerns our perceptions of our inner emotional world. An alternative tag for the same construct is trait emotional self-efficacy. People with high EI rankings believe they are "in touch" with their feelings and can regulate them in a way that promotes prosperity. These people must enjoy higher levels of happiness. The trait EI feature sampling domain aims to provide complete coverage of emotional aspects of personality. Trait EI rejects the idea that emotions can be artificially objectified in order to be graded accurately along the IQ lines (Petrides et al. 2007). The adult sampling domain of trait EI contains 15 facets: Adaptability, Assertiveness, Emotion perception (self and others), Emotion expression, Emotion management (others), Emotion regulation, Impulsiveness (low), Relationships, Self-esteem, Self-motivation, Social awareness, Stress management, Trait empathy, Trait happiness, Trait optimism (Petrides, 2010).
Research on emotional intelligence has been divided into two distinct areas of perspectives in terms of conceptualizing emotional competences and their measurements. On the one hand there is the ability EI model (Mayer et al. 1995) and on the other hand there is the trait EI (Petrides et al. 2001). Research evidence has consistently supported this distinction by revealing low correlations between the two (O'Connor et al. 2003; Warwick et al. 2004; Pérez et al. 2005; Mikolajczak et al. 2006; Brackett et al. 2011).”
As far as the possible links between our model and the Ability and Trait one are concerned we added the following sentences in chapter 4. The Pyramid of Emotional Intelligence: The Nine—Layer Model (lines 241-246):
“Our model includes some features from both constructions (the Ability EI and the Trait EI model) in a more hierarchical structure. The ability level refers to the ability to awareness (self and social) and management. The level of trait refers to the mood associated with emotions, and the tendency to behave in some way in emotional states considering and other important elements that this construction includes.”
2) Authors mention a potential link between EI and decision-making processes, but it seems unexplored. They should add some references concerning literature about EI and decision-making processes (e.g., Panno et al., https://doi.org/10.1027/1864-9335/a000247; Sevdalis et al., https://doi.org/10.1016/j.paid.2006.10.012 ). As the DM processes are higher-order cognitive factors that strongly influence daily life outcomes, the manuscript would benefit from adding a paragraph linking these aspects.
Reply: Thank you for providing these insights and we have added the following sentences in Discussion-Conclusion (lines 446-451):
“Research has shown this positive correlation between emotional intelligence and cognitive processes, and this demonstrates the important role that emotional intelligence plays with emotion and cognition, thus empowering individuals and their personality and benefitting the whole society (Killgore et al. 2007; Sevdalis et al. 2007; Panno, 2016; Rezaeian et al. 2017; Megías et al. 2017).”
3)Finally, the manuscript would benefit from adding a section describing potential applied interventions of the model. Moreover, some suggestions on how people EI could be enhanced based on this model are needed. The discussion section of the manuscript reports a very small number of prompts and suggestions about applied research and future studies investigating EI phenomenon.
Reply: We agree and have added the following sentences to the Discussion-Conclusions (lines 445-456):
“The model of emotional intelligence has been created with a better distinct classification. It is a more structured evaluation and intervention model with hierarchical levels to indicate each time the level of emotional intelligence that everyone is and by operation procedures to contribute to the strengthening of that level and progressive development of the individual to the next levels of emotional intelligence. It is a methodology for further development and evolution of the individual. This model can have applications as an evaluation, assessment and training tool in any aspect of life such as interpersonal relationships, work, health, special education, general education, academic success. Researchers claim that an emotional mind is important for a good life as much as an intelligent mind and in cases it matters more (Nelson et al. 2006). The ultimate goal should be to develop Emotional Intelligence, do further research on the benefits of such an important capacity and the correlations between layered Emotional Intelligence model and other variables.”
Minor Changes:
Row 195: there is a typo.
Reply: Typo was corrected.
Row 272: “clearly clarified” does not sound good.
Reply: The word “clearly” was deleted from the sentence as it does not sound good.
Rows 411/412: a reference is missing.
Reply: References were added.
Row 419: see above.
Reply: References were added.
Reviewer 3 Report
Thank you for presenting this work on emotional intelligence. Levels of awareness or levels through which to progress to acquire EI may be of interest to readers, however, the model may be founded on faulty reasoning. For example, I did not see literature differentiating trait and ability EI, the ways the two might interface, or discussion about whether this model would work within either or both of those constructs.
The paper would benefit from close editing and the disciplined use of APA documentation. For example, entire paragraphs were written with only one source noted at the end. Many sentences have no reference attached. At times the paper appears to be an opinion-based paper and at other times, a literary review. Because of the loose interpretation, EI is sometimes discussed inaccurately (for example, what has the criticism been for Goleman's claims that EI is more effective than intellect?). Are the criticisms valid? The paper would be strengthened by clearly delineating the style of writing and by eliminating summaries that reflect opinion rather than past findings to draw in the reader to view the writing as credible.
The development of a model requires more rigor in both the review of literature and the description of how the model was synthesized. In general, the paper would be strengthened by reducing the redundancy in descriptions, increasing the clarity of concepts related to the ways EI has been shaped over the years, and using discipline in writing to present a more cohesive narrative.
Author Response
We would like to thank the reviewers for careful and thorough reading of this manuscript and for the thoughtful comments and constructive suggestions, which help to improve the quality of this manuscript. With this cover letter, we will submit the revised manuscript (behavsci-270715) entitled, “A New Layered Model on Emotional Intelligence” for publication in Behavioral Sciences. Based the comments from the reviewers, we have made changes of the manuscript, which are detailed below (the reviewer’s comments are in italics). Changes made in the manuscript are marked using track changes and hope to meet with your approval.
REVIEWER #3:
Major changes:
1)Thank you for presenting this work on emotional intelligence. Levels of awareness or levels through which to progress to acquire EI may be of interest to readers, however, the model may be founded on faulty reasoning. For example, I did not see literature differentiating trait and ability EI, the ways the two might interface, or discussion about whether this model would work within either or both of those constructs.
Reply: We agree with the reviewer and have added the following paragraphs in chapter 3.3. Emotional Intelligence (lines 206-225):
“Petrides and Furnham (2001) developed the Trait Emotional Intelligence model which is a combination of emotionally-related self-perceived abilities and moods that are at the lowest levels of personality hierarchy evaluated through questionnaires and rating scales (Petrides & Furnham, 2001). The trait EI essentially concerns our perceptions of our inner emotional world. An alternative tag for the same construct is trait emotional self-efficacy. People with high EI rankings believe they are "in touch" with their feelings and can regulate them in a way that promotes prosperity. These people must enjoy higher levels of happiness. The trait EI feature sampling domain aims to provide complete coverage of emotional aspects of personality. Trait EI rejects the idea that emotions can be artificially objectified in order to be graded accurately along the IQ lines (Petrides et al. 2007). The adult sampling domain of trait EI contains 15 facets: Adaptability, Assertiveness, Emotion perception (self and others), Emotion expression, Emotion management (others), Emotion regulation, Impulsiveness (low), Relationships, Self-esteem, Self-motivation, Social awareness, Stress management, Trait empathy, Trait happiness, Trait optimism (Petrides, 2010).
Research on emotional intelligence has been divided into two distinct areas of perspectives in terms of conceptualizing emotional competences and their measurements. On the one hand there is the ability EI model (Mayer et al. 1995) and on the other hand there is the trait EI (Petrides et al. 2001). Research evidence has consistently supported this distinction by revealing low correlations between the two (O'Connor et al. 2003; Warwick et al. 2004; Pérez et al. 2005; Mikolajczak et al. 2006; Brackett et al. 2011).”
As far as the possible links between our model and the Ability and Trait one are concerned we added the following sentences in chapter 4. The Pyramid of Emotional Intelligence: The Nine—Layer Model (lines 241-246):
“Our model includes some features from both constructions (the Ability EI and the Trait EI model) in a more hierarchical structure. The ability level refers to the ability to awareness (self and social) and management. The level of trait refers to the mood associated with emotions, and the tendency to behave in some way in emotional states considering and other important elements that this construction includes.”
We appreciate the comments from the reviewers. We have tried to address the reviewers’ concerns in a proper way and believe that our paper has improved considerably. We would be happy to make further corrections if necessary and look forward to hearing from you soon.
Round 2
Reviewer 2 Report
I appreciated efforts by which authors addressed points raised in the previous round of revision.
As I suggested in the previous round of revision, authors should also refer to this work (Panno, A., Donati, M.A., Chiesi, F., & Primi, C. (2015). Trait emotional intelligence is related to risk-taking through negative mood and anticipated fear. Social Psychology, 46, 361-367. http://dx.doi.org/10.1027/1864-9335/a000247), when linking the emotional intelligence (EI) to decision-making processes. It sheds light on mechanisms underlying the relationship between EI and risk-taking. In particular, it shows how EI is related to risk-taking through both negative mood and anticipated fear, i.e. two factors naturally occurring in decision-making processes. Thus, it has been pointed out that EI can regulate emotions engendered from daily circumstances. This is relevant because such an aspect of EI has a number of effects on people daily life (e.g., reducing bias, downregulating negative affect and supporting more rational decisions).
I am not saying that authors have to drop the reference Panno 2016 that they cite in the latest version of the manuscript, but I think, authors can refer to Panno et al., 2015 to highlight how an enhanced EI self-efficacy can improve individuals daily life by downregulating negative affect that biases decision-making processes.
Author Response
REVIEWER #2:
Major changes:
1)As I suggested in the previous round of revision, authors should also refer to this work (Panno, A., Donati, M.A., Chiesi, F., & Primi, C. (2015). Trait emotional intelligence is related to risk-taking through negative mood and anticipated fear. Social Psychology, 46, 361-367. http://dx.doi.org/10.1027/1864-9335/a000247), when linking the emotional intelligence (EI) to decision-making processes. It sheds light on mechanisms underlying the relationship between EI and risk-taking. In particular, it shows how EI is related to risk-taking through both negative mood and anticipated fear, i.e. two factors naturally occurring in decision-making processes. Thus, it has been pointed out that EI can regulate emotions engendered from daily circumstances. This is relevant because such an aspect of EI has a number of effects on people daily life (e.g., reducing bias, downregulating negative affect and supporting more rational decisions).
I am not saying that authors have to drop the reference Panno 2016 that they cite in the latest version of the manuscript, but I think, authors can refer to Panno et al., 2015 to highlight how an enhanced EI self-efficacy can improve individual’s daily life by downregulating negative affect that biases decision-making processes.
Reply: We appreciate this observation and we have added the following sentences to the Discussion-Conclusions (lines 447-449):
“Results show that negative mood and anticipated fear are two factors of the relationship between trait EI and risk-taking in decision-making processes among adults (Panno et al. 2015).”
Reviewer 3 Report
It is apparent that work has been done to improve the article and a significant amount of new information has been added to support the ways our thinking about intelligence and EI has been shaped. The article would benefit from clearly stating its design. For example, is the model founded upon a literature review, and if so, perhaps a more suitable design would include a comprehensive and rigorous review of the literature such as through systematic review or meta-analysis.
Great care must be taken to give credit to authors for all information (reference above to "ethical concerns"). The paper includes paragraphs (see lines 19-26 & 44-56) that lack references. Take care to include references in summary statements, such as lines 158-163 (combine into one paragraph and provide references throughout).
In addition to inserting references throughout the paper as needed, punctuation, word choice and verb tense require close attention, 221-222. Some examples are found on the following lines: 20 (no hyphen needed), 23 (think and behavior/ verb tense), line 25 (eliminate sentence starting with "People pay attention...", need to reword lines 27, 44-56 & 61-62,158-163, etc.
The paper may be more readable if the lists of bullets are eliminated. Think about the lists of terms that make up various theories and try to summarize the information differently. There continues to be redundancy in defining EI over and over in various ways such as in the introduction followed by lines 164, 226, 236, etc
Last, the idea of the model is intriguing. More work must be done to show the processes involved with moving from one level to the next. Perhaps there are other tables or charts that could illustrate the movement between levels and the processes involved more clearly than by using narrative alone.
Author Response
REVIEWER #3:
Major changes:
1)Great care must be taken to give credit to authors for all information (reference above to "ethical concerns"). The paper includes paragraphs (see lines 19-26 & 44-56) that lack references. Take care to include references in summary statements, such as lines 158-163 (combine into one paragraph and provide references throughout).
Rows 19-26: These rows were reworded, and references were added:
“Emotional intelligence is of great interest to scientists and researchers and studies from the past until nowadays continue to be made about the nature of emotional intelligence, its measurement, structure, positive or negative effects and its relationship with many research fields (García-Sancho et al. 2014; Costa et al. 2015; Cabello et al. 2016; Mayer et al. 2016; Petrides et al. 2016; Naseem, 2017). Its influence in daily life in short and long term is important.”
“García-Sancho, E., Salguero, J. M., & Fernández-Berrocal, P. (2014). Relationship between emotional intelligence and aggression: A systematic review. Aggression and violent behavior, 19(5), 584-591.
Costa, A., & Faria, L. (2015). The impact of emotional intelligence on academic achievement: A longitudinal study in Portuguese secondary school. Learning and Individual Differences, 37, 38-47.
Cabello, R., Sorrel, M. A., Fernández-Pinto, I., Extremera, N., & Fernández-Berrocal, P. (2016). Age and gender differences in ability emotional intelligence in adults: A cross-sectional study. Developmental psychology, 52(9), 1486.
Mayer, J. D., Caruso, D. R., & Salovey, P. (2016). The ability model of emotional intelligence: Principles and updates. Emotion Review, 8(4), 290-300.
Petrides, K. V., Mikolajczak, M., Mavroveli, S., Sanchez-Ruiz, M. J., Furnham, A., & Pérez-González, J. C. (2016). Developments in trait emotional intelligence research. Emotion Review, 8(4), 335-341.
Naseem, K. (2017). Job Stress and Employee Creativity: The mediating role of Emotional Intelligence. International Journal of Management Excellence, 9(2), 1050-1058.”
Rows 44-56: References to support our sayings were added:
“Heim, M. (1995). The design of virtual reality. Body & Society, 1(3-4), 65-77.
Sternberg, R. J. (Ed.). (2000). Handbook of intelligence. Cambridge University Press.
Mackintosh, N. J. (2011). IQ and human intelligence. Oxford University Press.
Eysenck, H. (2018). Intelligence: A new look. Routledge.”
Rows 158-163: References were added:
“Fredrickson, B. L. (2001). The role of positive emotions in positive psychology: The broaden-and-build theory of positive emotions. American psychologist, 56(3), 218.
Cohn, M. A., Fredrickson, B. L., Brown, S. L., Mikels, J. A., & Conway, A. M. (2009). Happiness unpacked: positive emotions increase life satisfaction by building resilience. Emotion, 9(3), 361.
Pekrun, R., Muis, K. R., Frenzel, A. C., & Götz, T. (2017). Emotions at school. Routledge.
Ruvalcaba-Romero, N. A., Fernández-Berrocal, P., Salazar-Estrada, J. G., & Gallegos-Guajardo, J. (2017). Positive emotions, self-esteem, interpersonal relationships and social support as mediators between emotional intelligence and life satisfaction. Journal of Behavior, Health & Social Issues, 9(1), 1-6.
Goleman, D. P. (1995). Emotional intelligence: Why it can matter more than IQ for character, health and lifelong achievement.
Ciarrochi, J., Forgas, J. P., & Mayer, J. D. (Eds.). (2001). Emotional intelligence in everyday life: A scientific inquiry. Psychology Press.
Coetzer, G. H. (2016). Emotional versus Cognitive Intelligence: Which is the better predictor of Efficacy for Working in Teams?. Journal of Behavioral and Applied Management, 16(2).”
Mestre, J. M., MacCann, C., Guil, R., & Roberts, R. D. (2016). Models of cognitive ability and emotion can better inform contemporary emotional intelligence frameworks. Emotion Review, 8(4), 322-330.
Mayer, J. D. (2001). Emotion, intelligence, and emotional intelligence.
Mitchell, R. L., & Phillips, L. H. (2015). The overlapping relationship between emotion perception and theory of mind. Neuropsychologia, 70, 1-10.
2)In addition to inserting references throughout the paper as needed, punctuation, word choice and verb tense require close attention, 221-222. Some examples are found on the following lines: 20 (no hyphen needed), 23 (think and behavior/ verb tense), line 25 (eliminate sentence starting with "People pay attention...", need to reword lines 27, 44-56 & 61-62,158-163, etc.
Reply: We apologize for these errors, and we have corrected the text as suggested.
Rows 221-222: “There is the ability EI model (Mayer et al. 1995) and the trait EI (Petrides et al. 2001).”
Row 20: Hyphen deleted.
Row 23: “…as a guide for their own thoughts and behaviors and for others in general.”
Row 25: Sentence was eliminated.
Row 27: “Intellectual ability is significant to succeed in everyday life in many and different sectors. But there are some other important components too that contribute to… “.
Rows 44-56: We reword line 52: “Intelligence is an important aspect of the mind that includes a lot of cognitive abilities…”
Rows 61-62: “The results in all trials correlated positively, Spearman believed, underlying the importance of general intelligence.”
Rows 158-163: “Research has shown the important role that emotions play in our lives in many fields (Fredrickson, 2001; Cohn et al. 2009; Pekrun et al. 2017; Ruvalcaba-Romero et al. 2017). Researchers have found that Emotional Intelligence is equal to or sometimes much more important than I.Q (Goleman, 1995; Ciarrochi et al. 2001; Coetzer, 2016). Emotion and intelligence are heavily linked (Mayer, 2001; Mitchell et al. 2015; Mestre et al. 2016). If you are aware of your own and others feelings, that will help you manage behaviors and relationships and predict success in many sectors.”
3)The paper may be more readable if the lists of bullets are eliminated. Think about the lists of terms that make up various theories and try to summarize the information differently. There continues to be redundancy in defining EI over and over in various ways such as in the introduction followed by lines 164, 226, 236, etc
Reply: Thank you for your suggestion.
We deleted bullets in lines 107-117, 117-128
Line 235-236 was deleted because of repetition of defining EI.
4)Last, the idea of the model is intriguing. More work must be done to show the processes involved with moving from one level to the next. Perhaps there are other tables or charts that could illustrate the movement between levels and the processes involved more clearly than by using narrative alone.
Reply: We have reflected this suggestion by adding a small chapter and by making another one pyramid where we have added the processes.
4. Cognitive and Metacognitive Processes in Emotional Intelligence Model
Cognition encompasses processes such as attention, memory, evaluation, problem solving language and perception (Best, 1999; Coren, 2003). Cognitive processes use existing knowledge and generate new knowledge. Metacognition is defined as the ability to monitor and reflect upon one’s own performance and capabilities (Flavell, 1979; Dunlosky et al. 2009). It is the ability of individuals to know their own cognitive functions in order to monitor and control their learning process (Cox, 2005; Caro Pineres et al. 2013). The idea of meta-cognition relies on the distinction between two types of cognitions: primary and secondary (McGuire et al. 2014). Metacognition includes a variety of elements and skills such as Metamemory, Self-Awareness, Self-Regulation and Self-Monitoring (Vockel, 2009; Caro Pineres et al. 2013).
Metacognition in Emotional Intelligence means that an individual perceives his/her emotional skills (Elipe et al. 2015). Its processes involve emotional-cognitive strategies such as awareness, monitoring and self-regulation (Wheaton, 2012). Apart from the primary emotion a person can experience and direct thoughts that accompany this emotion, people may have additional cognitive functions that monitor a given emotional situation (Scheir et al. 1982), evaluate the relationship between emotion and judgment (Mayer et al. 1985), try to manage their emotional reaction (Isen, 1984) for the improvement of their own personality and that will motivate them help other people for better interpersonal interactions. Applying metaknowledge to socio-emotional contexts should lead to the opportunity to learn to correct one’s emotional errors and to promote the future possibility of a proper response to the situation while maintaining and cultivating the relationship (Kelly et al. 2011).
In the pyramid of Emotional Intelligence that we presented in order to move from one layer to another to reach the higher levels, cognitive and metacognitive processes are occurred (Figure 2).
Fig. 2 Cognitive and Metacognitive processes to move from a layer to another.
Again, thank you for giving us the opportunity to strengthen our manuscript with your valuable comments and queries. We have worked hard to incorporate your feedback and hope that these revisions persuade you to accept our submission.
Round 3
Reviewer 3 Report
Thank you for making the revisions. There are some additional suggestions below:
Lines 29, 32- references?
34: Speaks TO
46-HAVE been (state it again)
54-56- add this one-sentence paragraph to section above
60- need date after Spearman's name.
60-70: put references at end of each sentence
73: centered
158-159: source?
160-165: Please specify if talking about Trait or Ability EI
196-201: Four elements (5 are listed)
233: taking INTO consideration
238: ability OF
242: add this one sentence paragraph to paragraph above it
246-250: source? check grammar
436-437: reword
449: contributes TO
453: human lesions?
463-464: talk about Trait or EI ability
Author Response
We appreciate the interest that the editor and reviewers have taken in our manuscript and the constructive criticism they have given. We have addressed the minor changes of the reviewer.
REVIEWER #3
Minor Changes:
Thank you for your review of our paper. We have answered each of your points below.
1)Lines 29, 32- references?
Reply: “ Sternberg, R. J. (1997). The concept of intelligence and its role in lifelong learning and success. American psychologist, 52(10), 1030.
Busato, V. V., Prins, F. J., Elshout, J. J., & Hamaker, C. (2000). Intellectual ability, learning style, personality, achievement motivation and academic success of psychology students in higher education. Personality and Individual differences, 29(6), 1057-1068.
Kanazawa, S. (2004). General intelligence as a domain-specific adaptation. Psychological review, 111(2), 512.
Strenze, T. (2007). Intelligence and socioeconomic success: A meta-analytic review of longitudinal research. Intelligence, 35(5), 401-426.
McGrew, K. S. (2009). CHC theory and the human cognitive abilities project: Standing on the shoulders of the giants of psychometric intelligence research.
Carroll, J. B. (1993). Human cognitive abilities: A survey of factor-analytic studies. Cambridge University Press.
Gendron, B. (2004). Why emotional capital matters in education and in labour? Toward an optimal exploitation of human capital and knowledge management.
Di Fabio, A. (Ed.). (2011). Emotional intelligence: New perspectives and applications. InTech.”
2)34: Speaks TO
Reply: Done.
3)46-HAVE been (state it again)
Reply: Done.
4)54-56- add this one-sentence paragraph to section above
Reply: Done.
5)60- need date after Spearman's name.
Reply: Done.
6)60-70: put references at end of each sentence
Reply: “Spearman, C. (1914). The theory of two factors. Psychological Review,21(2), 101.
Duncan, J., Seitz, R. J., Kolodny, J., Bor, D., Herzog, H., Ahmed, A., ... & Emslie, H. (2000). A neural basis for general intelligence. Science,289(5478), 457-460.
Spearman, C. (1927). The measurement of intelligence. Nature, 120(3025), 577.”
7)73: centered
Reply: Done.
8)158-159: source?
Reply: “Brackett, M. A., Rivers, S. E., & Salovey, P. (2011). Emotional intelligence: Implications for personal, social, academic, and workplace success. Social and Personality Psychology Compass, 5(1), 88-103.
Libbrecht, N., Lievens, F., Carette, B., & Côté, S. (2014). Emotional intelligence predicts success in medical school. Emotion, 14(1), 64.
Rezvani, A., Chang, A., Wiewiora, A., Ashkanasy, N. M., Jordan, P. J., & Zolin, R. (2016). Manager emotional intelligence and project success: The mediating role of job satisfaction and trust. International Journal of Project Management, 34(7), 1112-1122.”
9)160-165: Please specify if talking about Trait or Ability EI
Reply: Done.
10)196-201: Four elements (5 are listed)
Reply: Corrected.
11)233: taking INTO consideration
Reply: Done.
12)238: ability OF
Reply: Done.
13)242: add this one sentence paragraph to paragraph above it
Reply: Done.
14)246-250: source? check grammar
Reply: References were added, and grammar was checked.
“Mayer, J. D., DiPaolo, M., & Salovey, P. (1990). Perceiving affective content in ambiguous visual stimuli: A component of emotional intelligence. Journal of personality assessment, 54(3-4), 772-781.
Vuilleumier, P. (2005). How brains beware: neural mechanisms of emotional attention. Trends in cognitive sciences, 9(12), 585-594.
Brosch, T., Pourtois, G., & Sander, D. (2010). The perception and categorisation of emotional stimuli: A review. Cognition and emotion, 24(3), 377-400.
Isomura, T., & Nakano, T. (2016). Automatic facial mimicry in response to dynamic emotional stimuli in five-month-old infants. Proc. R. Soc. B, 283(1844), 20161948.”
15)436-437: reword
Reply: Done.
16)449: contributes TO
Reply: Done.
17)453: human lesions?
Reply: Deleted.
18)463-464: talk about Trait or EI ability
Reply: Done.
We hope the revised version is now suitable for publication and look forward to hearing from you.
Best regards
Athanasios Drigas
Chara Papoutsi